# All Fiber-Optic Immunosensors Based on Elliptical Core Helical Intermediate-Period Fiber Grating with Low-Sensitivity to Environmental Disturbances

**DOI:** 10.3390/bios12020099

**Published:** 2022-02-06

**Authors:** Junlan Zhong, Shen Liu, Tao Zou, Wenqi Yan, Min Zhou, Bonan Liu, Xing Rao, Ying Wang, Zhongyuan Sun, Yiping Wang

**Affiliations:** 1Key Laboratory of Optoelectronic Devices and Systems of Ministry of Education/Guangdong Province, College of Physics and Optoelectronic Engineering, Shenzhen University, Shenzhen 518060, China; zhongjunlan@email.szu.edu.cn (J.Z.); szu_zoutao@163.com (T.Z.); yanwenqi2020@email.szu.edu.cn (W.Y.); minzhou2020@163.com (M.Z.); ssamliu@163.com (B.L.); 2060453012@email.szu.edu.cn (X.R.); yingwang@szu.edu.cn (Y.W.); sunzhongyuan@hotmail.co.uk (Z.S.); ypwang@szu.edu.cn (Y.W.); 2Shenzhen Key Laboratory of Photonic Devices and Sensing Systems for Internet of Tings, Guangdong and Hong Kong Joint Research Centre for Optical Fibre Sensors, Shenzhen University, Shenzhen 518060, China

**Keywords:** elliptical core helical intermediate-period fiber grating, immunosensor, human immunoglobulin, all fiber-optic sensor, insensitivity to environmental disturbances

## Abstract

An all fiber-optic immunosensor based on elliptical core helical intermediate-period fiber grating (E-HIPFG) is proposed for the specific detection of human immunoglobulin G (human IgG). E-HIPFGs are all-fiber transducers that do not include any additional coating materials or fiber architectures, simplifying the fabrication process and promising the stability of the E-HIPFG biosensor. For human IgG recognition, the surface of an E-HIPFG is functionalized by goat anti-human IgG. The functionalized E-HIPFG is tested by human IgG solutions with a concentration range of 10–100 μg/mL and shows a high sensitivity of 0.018 nm/(μg/mL) and a limit of detection (LOD) of 4.7 μg/mL. Notably, the functionalized E-HIPFG biosensor is found to be insensitive to environmental disturbances, with a temperature sensitivity of 2.6 pm/°C, a strain sensitivity of 1.2 pm/με, and a torsion sensitivity of −23.566 nm/(rad/mm). The results demonstrate the considerable properties of the immunosensor, with high resistance to environmental perturbations, indicating significant potential for applications in mobile biosensors and compact devices.

## 1. Introduction

Human immunoglobulin G (IgG) is a kind of large molecule in blood, with a molecular weight of roughly 150 kD and a tetrameric quaternary structure consisting of four peptide chains: two heavy chains (HCs) and two light chains (LCs) [1,2]. According to a large body of research, IgGs play a critical role in defending against viruses and human health-maintaining [3,4,5,6]. For example, various IgGs can pass across the placental barrier and act as anti-infection immunity for newborns [7,8]. Furthermore, the fragment crystallizable region (known as Fc receptors) of an IgG interacts with cell surface receptors, which allows the antibody to activate the immune system. The Fc region of some IgG subclasses can connect to protein A of Staphylococcus aureus bacteria, allowing for antibody purification and immunodiagnosis [9]. Therefore, a stable and easy-to-handle IgG sensor is vital in human healthcare monitoring, disease diagnosis, and therapeutic research. The most common commercially available technology for IgG detection is enzyme-linked immunosorbent assay (ELISA). An LOD of 0.06 µg/mL for IgG measurement was reported in 2002 [10]. Other methods include immunochemical biosensor and quartz crystal microbalances, with the LOD reaching 49 ng/mL and 5 µg/mL, respectively [11,12]. Recently, a diffraction grating immunosensor was proposed for specific detection with the LOD of 1.3 × 10^−^^8^ M [13]. However, these methods are either time-consuming or complex processes. 

To address the limitations of conventional techniques for immunosensing, optical fiber sensors, including interferometric fiber-optic sensors [14], microfiber Bragg gratings [15,16,17,18], tilted fiber Bragg gratings (TFBG) [19], surface polarized resonance (SPR) [20,21], and long-period fiber gratings (LPFG) [22,23], were proposed as a biophotonic platform for antibody–antigen interaction monitoring and specific biomolecules’ detection. Among them, LPFGs with various structures and surface modifications were of great interest as immunosensors due to their high sensitivity, outstanding mechanical structure, easy fabrication, good biocompatibility, and free-label detection [23,24,25,26,27,28,29]. For example, in 2010, Wang et al. utilized a fully distributed LPFG coated with a functional film as an immunosensor and demonstrated that this immunosensor could detect specific antigen–antibody binding without cross-sensitivity to nonspecific binding agents [27]. Pilla et al. reported a modified LPFG immunosensor with a high RI overlay adjusted to a high-sensitivity operating point. The bovine serum albumin (BSA) and glutaraldehyde were utilized to immobilize IgGs on the device surface for goat anti-human IgG antibody (anti-IgG) detection [28]. A deposited dual-peak LPFG with graphene oxide (GO) nanosheets has been proposed as a platform for immunosensing and experimentally proved that the LPFG-based sensor had an RI sensitivity of 2538 nm/RIR and ultrahigh anti-IgG detection sensitivity, with a 7 ng/mL detection limit [23]. Very recently, an IgG sensor consisting of a graphene oxide (GO)-coated-U-bent LPFG inscribed in a two-mode fiber (TMF) was reported with a low limit of detection (23 ng/mL) in the 3–20 μg/mL range [24]. However, for an immunoassay, most of these optical fiber biosensors need to be combined with extra surface materials and/or multi-fiber architectures, resulting in complicated fabrication procedures that affect stability and weaken coupling strength [21,29]. Furthermore, some of these sensors have a high susceptibility to outside interference, which can impact the stability and the accuracy of the immunosensor.

Helix intermediate-period fiber gratings (HIPFGs) are a type of period fiber grating with a helical grating structure whose period is less than 100 μm, with a grating length of only a few centimeters [30]. According to the papers discussed above, HIPFGs have been widely used as all-fiber sensors because they offer low insertion loss, excellent stability, and insensitivity to environmental parameters such as temperature, strain, and torsion, in addition to the benefits of typical LPFGs [30]. The following is a sensing principle description of the HIPFG-based immunosensor: the helical grating structure of HIPFGs introduces periodic perturbations to the refractive index (RI) in optical fiber and consequently contributes to mode coupling of the fundamental core mode into the forward propagation cladding mode. In the phase-match condition, the coupling strength becomes relatively strong and visible as resonance bands in the transmission spectrum. The *i*-th resonant wavelengths satisfy the following equation [30,31,32,33,34]:(1)$$\lambda_{i}=\left(n_{core}-n_{cladding}^{i}\right)\Lambda ,$$
where $n_{core}$ and $n_{cladding}^{i}$ are the effective refractive index of the fundamental core mode and the *i*th order cladding mode, respectively. $\lambda_{i}$ is the *i*th order resonance wavelength, and Λ is the grating period of the HIPFG. According to Equation (1), the resonant wavelength $\lambda_{i}$ is mainly perturbed by the evanescent field of the cladding mode; in other words, the change of the surrounding RI, which is attributed to the interaction of antibody and antigen. Therefore, by monitoring the resonant wavelength of the HILPG, the RI changes of the surrounding medium linked to the antibody–antigen binding condition can be observed.

In this work, we propose an all fiber-optic human IgG immunosensor by inscribing HIPFG on elliptical core fiber, which is named E-HIPFG here. Compared with an HIPFG inscribed in single-mode fiber with the same fabrication process, an E-HIPFG possesses higher coupling strength with a smaller sensing area, which improves the sensor resolution and allows micro-volume detection. This work is mainly divided into three parts. Firstly, an E-HIPFG sensor with high RI sensitivity is fabricated by a hydrogen–oxygen flame heating system and demonstrated by recording the transmission spectra of the E-HIPFG. Secondly, the E-HIPFG is functionalized by a chemical crosslinking method, and goat anti-human IgG is immobilized on it for specific human IgG detection. Lastly, the sensitivity and LOD for human IgG recognition of the E-HIPFG-based immunosensor are investigated by putting the sensor into human IgG solutions with a concentration range from 10 to 100 μg/mL. Significantly, the outstanding resistance to environmental disturbances of the functionalized E-HIPFG is further proved by its low temperature sensitivity, strain sensitivity, and torsion sensitivity.

## 2. Materials and Methods

### 2.1. Materials

The (3-aminopropyl) triethoxysilane (APTES, H_2_N(CH_2_)_3_Si(OC_2_H_5_)_3_, 99%) and glutaraldehyde (C_5_H_8_O_2_, 50% aqueous solution) were provided by Shanghai Aladdin Biochemical Technology Co., Ltd. (Shanghai China). The bovine serum albumin (BSA, pH 7), phosphate buffer saline (1 × PBS, pH 7.4), sodium hydroxide (NaOH), sulfuric acid (H_2_SO_4_), and hydrogen peroxide (H_2_O_2_) were purchased from Sigma-Aldrich (St. Louis, MO, USA). Human IgG and goat anti-human IgG antibodies (anti-IgG) were procured from Proteintech Group Inc. (Wuhan, China). Rabbit IgG was obtained from Bioworld Technology CO., Ltd. (Nanjing, China). All these reagents were used without further purification. The elliptical core fiber (P1C0027901B0, YOEC) utilized for the experiment was provided by Yangtze Optical Electronic Co., Ltd. (Wuhan, China). The fiber has a diameter of 125 μm, and the elliptical core has major and minor axis sizes of 10.0 and 5.0 μm, respectively.

### 2.2. Fabrication of the E-HIPFG

The performance of the investigated IgG immunosensor is significantly dependent on the use of high-quality E-HIPFG. In this work, the E-HIPFG was manufactured using a high-efficiency hydrogen–oxygen flame heating system that included a high-precision rotator, two translation stages, and a hydrogen generator, as schematically represented in Appendix A. The fabrication process is detailed in the Appendix A. Previous research has demonstrated that the manufacturing technique is a high-efficiency and quick approach for producing HIPFG with strong chemical–physical stability, high RI sensitivity, and repeatability [30,31,33]. The fabricated E-HIPFG grating period was determined to be 17.5 μm using the formula Ʌ = 30 V_2_/Ω, where Ω is the rotation rate of left translation stages, and V_2_ is the working velocity of another stage. The period of E-HIPFG was half of the period of HIPFG in single-mode fiber produced using the same system parameters [30]. The half grating period of the E-HIPFG is due to the two-fold axial symmetry of the elliptical core fiber, which possesses inherent major and minor axis structures. The smaller period of E-HIPFG leads to more periods of E-HIPFG than of HIPFG in the case of the same grating length. Consequently, it is easier for the E-HIPFG to reach the optimal coupling strength [35].

### 2.3. The Surface Functionalization of E-HIPFG

The surface of E-HIPFG must be modified by specific chemical groups to immobilize anti-IgG before it can be utilized as a label-free immunosensor for human IgG. This work employed the APTES as a chemical connector for anti-IgG immobilization, similar to former reports [20,36]. The surface functionalization strategy was separated into five steps, which are depicted graphically in Figure 1. Firstly, to remove the organic contaminants and activate negatively charged hydroxyl groups on the E-HIPFG surface, the E-HIPFG grating portion was bathed in a piranha solution (3:1; H_2_SO_4_/H_2_O_2_) for 30 min, followed by flushing it with deionized (DI) water thoroughly and drying it under a stream of nitrogen gas (Figure 1a). Secondly, the cleaned E-HIPFG was silanized with freshly produced APTES in ethanol (2% *v*/*v*) for 2 h, where the APTES interacted with the hydroxyl moieties to create amidogen groups on the E-HIPFG surface (Figure 1b). It was reported that immersing silicon substrate in the 2% APTES solution for 1 h is just enough to achieve monolayer APTES, which presents as a topologically uniform film on silicon substrate [36]. Then, the E-HIPFG was immediately rinsed by ethanol and baked at 110 °C for 30 min to improve the stability of the APTES layer [20,36]. Thirdly, the APTES-modified E-HIPFG was transferred into glutaraldehyde in PBS buffer (2.5% *v*/*v*) for 30 min to generate carboxyl groups (Figure 1c). Subsequently, the carboxyl-activated E-HIPFG was reacted with 100 μg/mL of goat anti-human IgG in PBS solution (Figure 1d). The success of anti-IgG immobilization is confirmed by the wavelength shift of the resonance dip, as shown in Appendix A. Finally, the E-HIPFG was submerged in 10 mg/mL BSA for 15 min to block residual binding sites (Figure 1e), after which, it was soaked in PBS buffer for 15 min when the transmission spectrum of E-HIPFG became steady. All the functionalization processes were carried out at room temperature.

### 2.4. Sensing System

The E-HIPFG sensor was launched by an amplified spontaneous light source (ASE, NKT Photonics, Birkerød, Denmark) and the transmission spectra were monitored by an optical spectrum analyzer (OSA, Yokogawa, Musashino, Tokyo, AQ6370C) with the wavelength cover from 1250 to 1650 nm and a resolution of 0.02 nm. To decrease the bend cross-sensitivity of the sensor, the two ends of the sensing area were set straight by two stages throughout the measurement process. The refractive index (RI) sensing characteristics of the E-HIPFG were estimated by sinking it into a series of glucose solutions held by a glass slide. The concentration range of glucose solutions was set at 0–200 mg/mL, and the corresponding RI range was 1.33–1.36, where the RI of protein solutions is frequently situated. After each measurement, the E-HIPFG was cleaned with DI water to remove leftover liquid.

The goat anti-human IgG-immobilized E-HIPFG was prewashed in PBS buffer for 10 min before being immersed in 200 μL of human IgG solutions with the concentration ranging from 10 to 100 μg/mL (Figure 1f). To take the measurement, the E-HIPFG sensor was immersed in human IgG solutions on glass slide. After 1 h of soaking in IgG solutions, the sensor system was used to monitor the transmission spectra of the E-HIPFG. The previously used IgG solution was extracted, and the glass dish was washed with DI water after each detection. A subsequent plunging of the E-HIPFG into 200 μL of NaOH solution (1 mM) for 15 min was conducted to dissociate the bonding antigen, followed by rinsing with PBS buffer (Figure 1g). The transmission spectra of rabbit IgG and BSA solutions with concentrations of 40 and 100 μg/mL were also recorded and compared with those of 40 and 100 μg/mL human IgG solutions to determine the selective detection of the immunosensor. The same DI water washing and NaOH-PBS rinsing were implemented after each detection. All these tests were conducted at room temperature.

## 3. Results

### 3.1. The Characteristic of Bare E-HIPFG

Figure 2a,b show the structure of the fabricated E-HIPFG observed by an optical microscope, which proves that periodic RI perturbations are induced in the core of fiber. Furthermore, the E-HIPFG has no fiber deformation, promising mechanical qualities and flexibility for the IgG immunosensor. The E-HIPFG length L is 2.088 mm, and the grating period is 17.5 μm. The cross-section view of the fabricated E-HIPFG is illustrated in Figure 2c. The diameters of the elliptical core fiber were changed to 9.67 and 5.39 μm, and the diameter reduced to 121 μm after the HIPFGs were inscribed in the elliptical core fiber. The original spectrum of the bare E-HIPFG measured from 1250 to 1650 nm, as illustrated in Figure 2d, with an insertion loss of 2.6 dB. Five discrete dips appear in the spectrum, and the details are listed in Table 1. Dip-1 and Dip-3 are single dips, whereas Dip-2, Dip-4, and Dip-5 present as dual dips. The dual-dip characteristic, similar to HIPFG, is attributed to a difference in the efficient RI of the cladding modes TM (transverse magnetic) and TE (transverse electric), which grows with the launch wavelength and cladding mode order [37,38].

The RI sensitivity of the E-HIPFG is evaluated by submerging it in a series of glucose solutions with an RI cover from 1.33 to 1.36, and the evolving spectra are exhibited in Figure 3a. As the RI increases, all the resonance dips shift to the longer wavelength side. The inset figure illustrates the enlarged view of the orange region (Dip-4) of the spectra in Figure 3a. The gray line represents the spectrum of the E-HIPFG with an RI of 1.33, where the resonance wavelength shifts to 1547.77 nm and the coupling strength decreases to −10.267 dB, comparing the corresponding features of the spectrum in air. The dual dip Dip-4 has emerged into a broad single-dip due to the increase in the surrounding RI, which subsequently resulted in a drop in the mode index difference of the TE and TM modes [25]. A similar dip emergence is also observed with Dip-5. We plot relationships between the dip wavelength shifts and the change of the surrounding IR by setting the corresponding values of 1.33 RI solution as a reference; see Figure 3b. As the IR increases, the dip wavelength shifts, giving linear relationships to RI with sensitivity and R2, as listed in Table 1. According to the data presented in Table 1, Dips 4 and 5, in particular, have strong coupling strengths and high RI sensitivities of 239.78 nm/RIU (RI unit) and 270.67 nm/RIU, respectively, which can improve the performance of the E-HIPFG-based immunosensor. Thus, the Dip-5 has influential noise owing to the low light intensity of ASE with a wavelength above 1600 nm. Therefore, in the following discussion, we will evaluate the detection performance of the E-HIPFG using Dip-4.

### 3.2. Immunosensing Properties of Functionalization E-HIPFG

The transmission spectra of the E-HIPFG at Dip-4 in PBS buffer and human IgG solutions with concentrations of 10, 20, 40, 60, 80, and 100 μg/mL were recorded and presented in Figure 4a to assess the sensing characteristics of the E-HIPFG biosensor for human IgG detection. Here, the PBS buffer is set as a reference sample for the IgG immunosensing. In contrast to the dip wavelength in PBS solution, redshifts occur with Dip-4 when the E-HIPFG is immersed in human IgG solutions, which is attributed to human IgG binding to the anti-IgG of E-HIPFG surface, consequently increasing the surrounding RI of the E-HIPFG sensor. The dip wavelength responses to the change of human IgG concentration are represented in Figure 4b. The blue dots represent the relationship between the dip wavelength shift and the target antigen concentration, and the blue solid line proves the satisfactory calibration curve (R_2_ = 0.9874). The sensor sensitivity of the base on Dip-4 is calculated to be 0.018 nm/(μg/mL) according to the slopes of fitting lines. The inset figure of Figure 4b presents the wavelength shifts of the blank sample in 1 h, according to which the wavelength shift ($\bar{∆\lambda_{black}}$) and the standard deviation (σ) were determined to be 0.02 nm and 0.022 nm, respectively. The LOD is calculated to be 4.7 μg/mL by the formula of $C_{LOD}=f^{-1}\left(\bar{∆\lambda_{black}}+3\sigma \right)$, described in previous research [17,39], where $f^{-1}$ is the inverse of the fitting function.

The specific detection for human IgG measurement is investigated by comparing the detection with rabbit IgG and BSA. Figure 5 presents the wavelength shifts when bathing the E-HIPFG sensor in human IgG solutions, rabbit IgG solutions, and BSA solutions with a 40 and 100 μg/mL concentration for one hour. The wavelength shifts are 0.69, 0.02, and 0.02 nm for human IgG, rabbit IgG, and BSA with a concentration of 40 μg/mL, respectively. Lastly, the corresponding values for the 100 μg/mL solutions are 1.84, 0.27, and 0.26 nm. The scale values measured in the rabbit IgG and BSA solutions are relatively lower than the values in the human IgG solution, indicating that the surface-functionalized E-HIPFG sensor can realize a selectivity detector for human IgG with low cross-sensitivity to other agents.

The cross-sensitivities of the E-HIPFG to external disturbances, including torsion, strain, and temperature, were also experimentally studied. To evaluate the torsion sensitivity, one end of the E-HIPFG was secured by a stationary stage, whereas the other end was fixed by a rotating stage, with a distance of 17 cm between the two stages. As the E-HIPFG is rotated clockwise and counterclockwise from 0 to ±360° with an interval of 60°, the wavelength shift of Dip-4 is observed and shown in Figure 6a. The wavelength transfers to the short-wavelength side with the E-HIPFG twist in the clockwise (CW) direction and shifts to the long-wavelength side in the counterclockwise (CCW) direction. The wavelength variation gives a considerable linear relationship with the torsion in both directions, and the torsion sensitivity (S_tor_) is calculated to be −23.566 nm/(rad/mm). Then, the strain sensitivity of the E-HLPFG is calculated by applying external strain from 0 to 2200 με with 100 με increments. A redshift happens on the dip wavelength of Dip-4, with the strain sensitivity (S_str_) estimated to be 1.2 pm/με; see Figure 6b. Lastly, the E-HIPFG was placed in a column oven with a temperature from 20 °C to 100 °C and a step of 10 °C to achieve its temperature sensitivity. The transmission spectra were reported after each temperature was kept constant. Figure 6c displays that the wavelength redshifts when the environmental temperature rises, and the temperature sensitivity (S_tem_) is determined to be 2.6 pm/°C according to the slope of the linear fit line.

We compare the IgG sensing sensitivity and anti-environmental interference capability of the proposed HIPFG immunosensor to other reported optical fiber sensors in Table 2. Although previous reports on IgG sensors are superior in LOD for IgG detection, most of them lack evaluation for environmental factors [40,41], and some of them are sensitive to those factors, which limits their application in mobile equipment and compact devices [24]. As listed in Table 2, the sensitivities for torsion, strain, and temperature of HIPFG are significantly smaller than the corresponding values of the previously reported optical fiber IgG sensor, providing stable and reliable sensing performance for IgG sensing. We attribute the robustness of the HIPFG to its compact structure with an easy fabrication method and without extra material or fiber restructuring.

## 4. Conclusions

We propose and implement a label-free immunosensor based on the all fiber-optic E-HIPFG for detecting human IgG, for the first time, as far as we know. The E-HIPFG that we have fabricated shows excellent mechanical behavior, flexibility, and a high refractive index (RI) of 239.78 nm/RIU. The E-HIPFG can be applied as an all-fiber immunosensor, guaranteeing the stability and reliability of IgG sensing. The E-HIPGH-based human IgG immunosensor achieves a high sensitivity of 0.018 nm/μg/mL and a LOD of 4.7 μg/mL with low cross-sensitivity to other agents. In particular, the investigated immunosensor proposes noteworthy resistance to ambient perturbations by giving a low-temperature sensitivity of 2.6 pm/°C, strain sensitivity of 1.2 pm/με, and a torsion sensitivity of −23.566 nm/(rad/mm), enabling robust and reliable sensing performance for human IgG detection. The suggested biosensor can be considered a prospective candidate component for medical diagnostics and clinical therapy due to its exceptional sensing properties.

## Figures and Tables

**Figure 1 biosensors-12-00099-f001:**
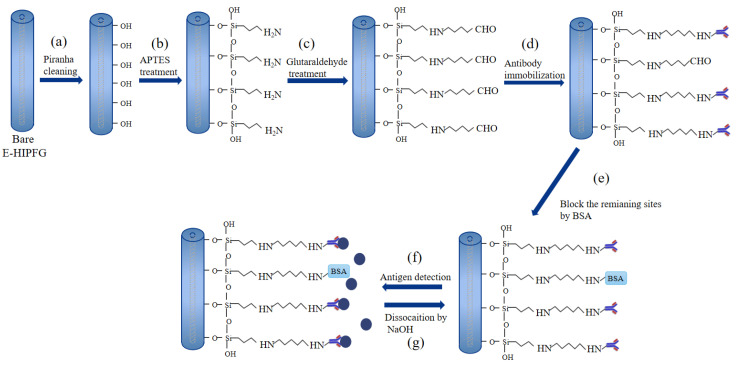
Schematic representation of the surface functionalization of E-HIPFG: (**a**) cleaning and activating the E-HIPFG by piranha solution; (**b**) generation of silane layer using APTES solution; (**c**) treatment with glutaraldehyde solution; (**d**) immobilization of anti-IgG; (**e**) blocking the remaining bonding sites by BSA solution; (**f**) human IgG detection and (**g**) dissociation of antibody–antigen bonding by NaOH.

**Figure 2 biosensors-12-00099-f002:**
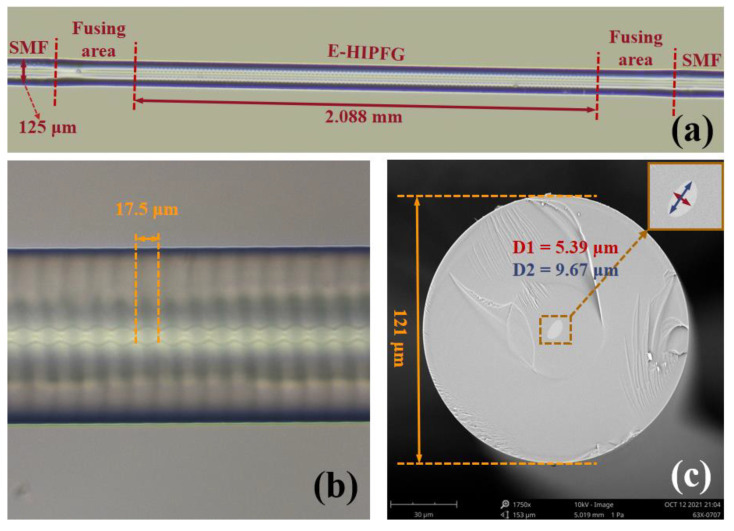
(**a**,**b**) Microscope imaging of the fabricated E-HIPFG with a of length 2.088 mm and a grating period of 17.5 μm. (**c**) The SEM of the cross-section of the fabricated E-HIPFG, the inset figure is the enlarged view of the elliptical core of the E-HIPFG. (**d**) The original spectrum of the bare E-HIPFG measured in air.

**Figure 3 biosensors-12-00099-f003:**
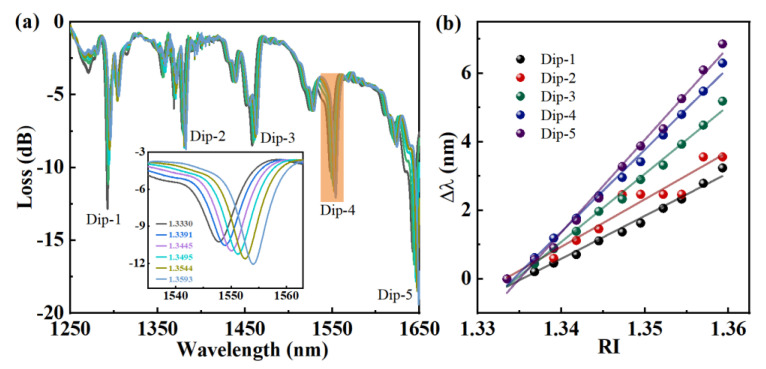
(**a**) The spectra of the E-HIPFG measured in glucose solutions with an RI range of 1.33–1.36. (**b**) The dependence of wavelength shifts on the RI for the sample.

**Figure 4 biosensors-12-00099-f004:**
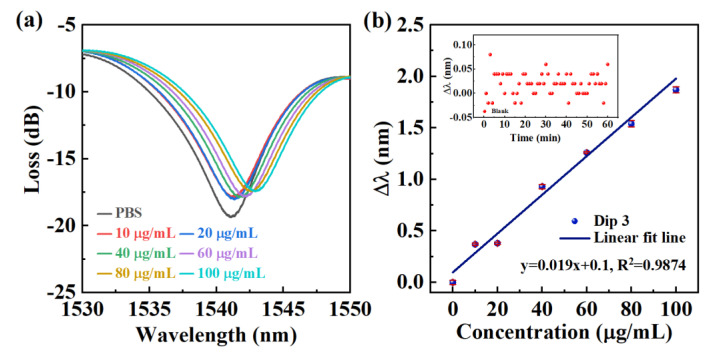
(**a**) The spectra of the functionalized E-HIPFG measured in human IgG solutions with concentrations range of 10–100 μg/mL; (**b**) the response of wavelength shift on antibody concentration. The inset figure of Figure 4b presents the wavelength shifts of the blank sample in 1 h.

**Figure 5 biosensors-12-00099-f005:**
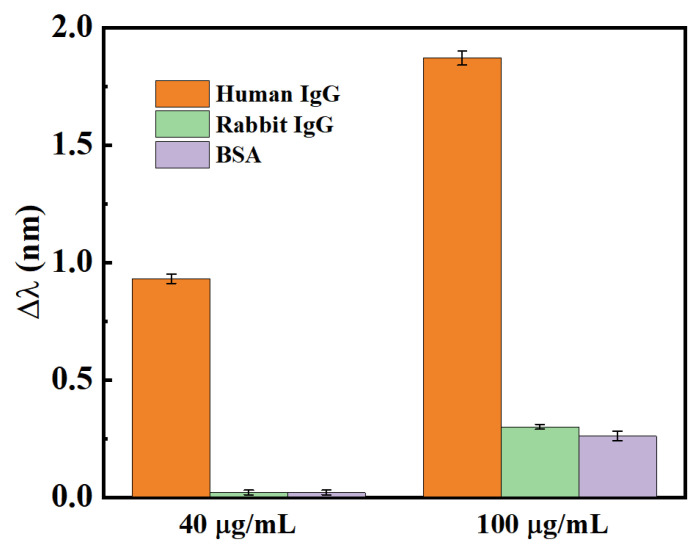
Specific detection for human IgG.

**Figure 6 biosensors-12-00099-f006:**
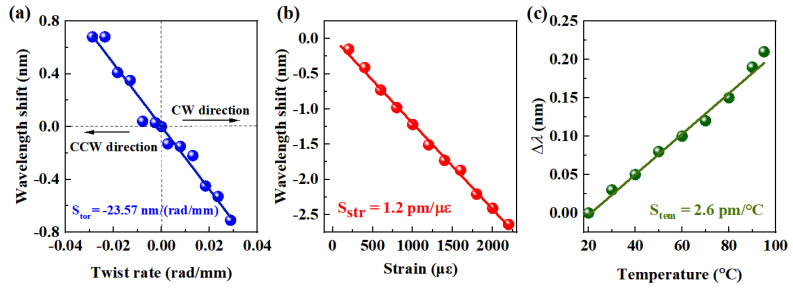
The Dip-4 of E-HIPFG responses to the change of torsion (**a**), strain (**b**) and temperature (**c**).

**Table 1 biosensors-12-00099-t001:** The dip features of the E-HIPFG.

Dip Features (in Air)	Dip-1	Dip-2	Dip-3	Dip-4	Dip-5
Wavelength (nm)	1238.31	1357.85	1439.72	1526.83	1619.67
Loss (dB)	−19.787	−22.164	−19.825	−23.621	−23.977
RI sensitivity (nm/RIU)/R^2^	125.16/0.9850	138.81/0.9524	198.76/0.9929	239.78/0.9948	270.67/0.9915

**Table 2 biosensors-12-00099-t002:** The comparison of the performances of various optical fiber IgG immunosensors.

Sensor Structure	Extra Materials/Architectures	S_IgG_ (nm/μg/mL)	LOD (μg/mL)	S_tor_ (nm/(rad/mm))	S_Str_ (pm/με)	Stem (pm/°C)	Ref.
U-bent LPFG	^1^ GO coating and U-bent structure	-	0.07	-	3.04	40.4	[13]
Dual-channel SPR	Au film/GO/Au^2^ NPs coating and MMF-PCF	-	0.015	-	-	5.1	[34]
S-Tapered Optical Fiber	None	-	0.028	-	-	−20	[33]
E-HIPFG	None	0.018	4.7	−23.566	1.2	2.6	This work

^1^ GO: graphene oxide; ^2^ NP: nanoparticles.

## Data Availability

Not applicable.

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
