# Peer review of "All Fiber-Optic Immunosensors Based on Elliptical Core Helical Intermediate-Period Fiber Grating with Low-Sensitivity to Environmental Disturbances"

_biosensors, 2022, doi:10.3390/bios12020099_

Round 1
Reviewer 1 Report
Optical fiber immunosensors are attracting extensive interest in recent years. This work reports a elliptical core helical short-period fiber grating based immunosensor for the specific determination of human IgG with the feature of mitigating the environmental interferences, such as temperature, torsion, and strain. The sensor can respond to the IgG with a density range from 10-100 μg/mL with a LOD of 1.1 μg/mL. This work is interesting and may contribute to the portable fiber biosensors. The manuscript writes clear and easy to read. However, several issues should be addressed before it can be accepted.
Major concerns:
- The major issue lies in the sensitivity and LOD of this sensor. The LOD is not very high in comparison with other type of fiber optic immunosensors. Is there any potential route for improving the sensitivity and LOD?
- What about the insertion loss of the fiber link after the elliptical core fiber was inserted and spliced with the SMFs?
- For a biosensor, the error bars which indicate the independent measurements should be included in the calibration curves -Fig. 4(b) and detection results-Fig.5. The error bars not only represent the test repeatablity but also are the important prerequisites for harvesting the real LOD and the signficance between the target and control. The related description can be found in “F. Chiavaioli, et al , Biosensors 2017, 7, 1.” (10.3390/bios7020023) , “Y. Ran, et al , Biosensors and Bioelectronics 2021, 179, 113081.” (10.1016/j.bios.2021.113081) and “A. Juste-Dolz, et al, Biosensors and Bioelectronics 2021, 176, 112916.” (10.1016/j.bios.2020.112916)
- The LOD calculation should be revised regarding those references above.
- Line 88-90, “By comparing with the HIPFG inscribed in single-mode fiber, E-HIPFG possesses higher coupling strength with smaller sensing area, which improve the sensor sensitivity and allow micro-volumes detection.” How to understand this argument? Please provide the explanation or the reference.
- Line 175-176 “After the NaOH-PBS-rinsing, the spectrum of the E-HIPFG was returned to the original one” To my knowledge, the sensor restoration using NaOH can not completely remove the immunobinding to the status before detection. Please double check the description.
Minor comments:
- Abstract, “018 nm/μg/ml”should read“0.018nm/(μg/ml)”
- Line 123,“É…=30 V2/Ω”, please add the notes of the “V” and “Ω”
- Line 142-143 “Then the E-HIPFG was immediately rinsed by ethanol and baked at 110°C for 30 minutes.” Why bake the E-HIPFG after silanization using APTES?
- Line 169, 173“200 μm”should read “200 μM”
- Line 172, what is“glass dish”and why use it in the experiment?
- Line 271-272 the test range“from 30OC to 9OC”is not agree with the temperature values in Fig.6c and the mark of “100 OC” is also missing in Fig.6c.
- The format of the Table 2 should be revised to show the items completely.
References suggestion:
An important type of fiber optic immunosensors is lacking in the overview of the state-of-the-art methods, i. e, the microfiber Bragg gratings,
For example,
Sridevi, et al, Biosensors and Bioelectronics 2015, 65, 251, (10.1016/j.bios.2014.10.033),
Juste-Dolz, et al, Biosensors and Bioelectronics 2021, 176, 112916. (10.1016/j.bios.2020.112916)
Ran, et al , Biosensors and Bioelectronics 2021, 179, 113081. (10.1016/j.bios.2021.113081)
Xiao, et al, Advanced Fiber Materials 2021. (10.1007/s42765-021-00087-7)
Reviewer 2 Report
In this work authors showed the use of a twisted long period grating for IgG detection. Authors have used the technology reported in previous publications to produce the intermediate long period grating and have functionalised the sensor in order to measure IgG. The work could be interesting for the readers, and taking into account the good literature review and results, my recommendation is to accept the paper after revising the following issues:
- Authors claim that the sensor has low cross sensitivity to external parameters. This is found even at the title of the manuscript. External parameters such as strain or torsion aren’t an issue if the sensor is kept fixed during the experiment. However, temperature can change a lot during one day and this could introduce errors on the measurement, even considering the low temperature sensitivity of 2.6 pm/ºC. Please comment on the possibility to solve this issue.
- Is the IgG concentration range of 10-100 μg/mL the one that we could find in human body? What is the minimum detectable by other technologies?
- The motivation behind the use of a E-HIPFG instead of standard long period gratings needs to be clarified.
- describe “Fc region” and “staphylocoglobulin A”.
- In line 89 authors said that the E-HIPFG possesses higher coupling strength. Describe the reason why these structures show higher coupling strength.
- In Line 123, describe the variables.
- Revise English, e.g. “…The i-th resonant wavelengths satisfying the following equation…”; “…wavelength in the transmission spectroscopy of the HILPG…”
Based on the above comments, my recommendation is minor revisions.
Round 2
Reviewer 1 Report
My concerns are well-addressed in the revised manuscript and it is ready for publication.